# A Simulator and First Reinforcement Learning Results for Underwater Mapping

**DOI:** 10.3390/s22145384

**Published:** 2022-07-19

**Authors:** Matthias Rosynski, Lucian Buşoniu

**Affiliations:** Department of Automation, Technical University of Cluj-Napoca, Memorandumului 28, 400114 Cluj-Napoca, Romania; lucian@busoniu.net

**Keywords:** underwater litter, mapping, AUV simulator, deep reinforcement learning

## Abstract

Underwater mapping with mobile robots has a wide range of applications, and good models are lacking for key parts of the problem, such as sensor behavior. The specific focus here is the huge environmental problem of underwater litter, in the context of the Horizon 2020 SeaClear project, where a team of robots is being developed to map and collect such litter. No reinforcement-learning solution to underwater mapping has been proposed thus far, even though the framework is well suited for robot control in unknown settings. As a key contribution, this paper therefore makes a first attempt to apply deep reinforcement learning (DRL) to this problem by exploiting two state-of-the-art algorithms and making a number of mapping-specific improvements. Since DRL often requires millions of samples to work, a fast simulator is required, and another key contribution is to develop such a simulator from scratch for mapping seafloor objects with an underwater vehicle possessing a sonar-like sensor. Extensive numerical experiments on a range of algorithm variants show that the best DRL method collects litter significantly faster than a baseline lawn mower trajectory.

## 1. Introduction

Rivers, lakes and seas are increasingly polluted by human-generated litter, which poses an increasing challenge to biodiversity preservation. One way to solve this problem efficiently is by using robots that can map and collect underwater waste, and the SeaClear project (https://seaclear-project.eu, accessed on 5 May 2022) aims to develop such a robotic team. This paper focuses on the mapping subproblem.

Classical mapping methods are nonadaptive, e.g., a lawn mower algorithm will perform a predefined trajectory consisting of a rectangular segments and will uniformly cover the entire map. As robotic systems, such as drones or unmanned underwater vehicles (UUVs), have limited runtime due, e.g., to battery capacity, and may be expensive in use, mapping must be performed efficiently and quickly. Formally, the uncertainty of the created map, measured, e.g., by the entropy, must be reduced as fast as possible. This is achieved by informative path planning (IPP) [1]. IPP methods are usually adaptive; they generate the path exploiting the map information accumulated from previous observations. One of the simplest such methods is the myopic algorithm, which chooses an action that maximizes the information gain [2] only at the next step.

This paper is the first to propose deep reinforcement learning (DRL) for long-horizon, nonmyopic IPP in underwater mapping. DRL is a sequential decision-making framework that maximizes a cumulative reward signal in problems with large-scale state signals, such as images, while the initial objective is to map litter, the DRL technique introduced works for mapping any quantity on the seafloor. A sonar-like sensor model is used, together with a 2.5D representation of the seafloor map in an occupancy grid format. The map height profile, belief and entropy are all part of the state signal plugged into a convolutional neural network.

Two DRL algorithms, double dueling deep Q-network (DDDQN) [3] and Rainbow [4], are applied to solve the mapping problem using a reward signal that encourages both discovering litter and reducing map entropy. Several mapping-specific improvements are made to the exploration schedule, to the prioritization of samples in the replay buffer [5] and by disallowing robot collisions with the seafloor or the limits of the operating area. Extensive experimental studies are performed where DRL is compared with the lawn mower strategy.

### 1.1. Related Work

Classical, collision-free planning [6,7] or shortest-path planning, such as the well-known Dijkstra’s method [8], do not work for IPP, since the dynamic evolution of the overall map must be considered instead of only the state of the robot.

Instead, in recent years, evolutionary algorithms became popular in mapping. Reference [9] introduced an evolutionary strategy for mapping cyanobacteria—an aquatic microorganism that performs photosynthesis. Instead of incrementally constructing a solution on a graph as classical planning would, [9] used a set of complete paths in continuous space together with an evolutionary strategy to maximize information gain. Furthermore, a replanning scheme is presented to adapt the path according to the measured data, allowing automatic focus on regions of interest, while ensuring that the robot sufficiently explores the area under study.

Another global evolutionary strategy for IPP was proposed by [10] for a drone that moves in 3D space and maps the terrain into a 2D grid representation. In some cases, the problem induces constraints on the starting and end position of the agent. Reference [11] proposed an evolutionary method for this case, which used an orienteering formulation for continuous-sampling sensors.

Since IPP can be formulated as a control problem, DRL is another good candidate to solve it, besides evolutionary methods. DRL for active mapping was used for Lidar (Light Detection and Ranging) sensors in [12]. There, however, the agent is moved along a known path and only the sensor itself is rotated on this path. Finding the optimal trajectory in IPP can also be stated as a graph search problem to be solved with RL [13]. Graph vertices store the position of the agent, and since this position is not a Markov state signal (because, as explained above, the map itself evolves dynamically), ref. [13] estimates Q-values using recursive neural networks where the input is the agent’s 2D position.

All the IPP problems above were solved with Gaussian process representations of the map. An alternative representation is the occupancy grid, where each grid cell is a Bernoulli distributed belief on the binary occupancy. This is more interesting from the DRL perspective because such a grid representation is analogous to the image inputs usually employed in DRL. In this context, ref. [14] compares a DRL agent with a myopic algorithm; DRL reaches slightly better results. The reward is the difference in entropy before and after making an observation (similar to the information gain). The objective is to reduce the entropy as fast as possible over the entire 2D environment with a 3×3 pixel sensor, thus, solving a coverage problem with obstacles.

### 1.2. Open Issues and Contribution

All the IPP approaches reviewed above have a key drawback: they cannot learn to focus on regions that are a priori more likely to hold information due to features in the environment, such as valleys where litter is likely to collect. A main question investigated here is therefore whether it is possible to develop a DRL mapping algorithm that learns to consider such features of the environment in order to find litter more efficiently. Indeed, the results reported in the sequel show that, with DRL, the agent can learn to exploit such a prior of the environment.

More generally speaking, no convincing application of DRL to 3D mapping has been reported. Therefore, a key objective of this paper is to develop such an application. Specifically, mapping is performed in a 3D environment with a 3D Bayesian sensor, using a 2.5D representation of the surface, while the motion of the agent is restricted to a 2D plane. Notably, due to the stochastic sensor, the problem is highly stochastic, which is known to be challenging for DRL [15].

Finally, to solve the mapping problem using DRL, a simple simulator is needed that cheap computationally. Existing UUV simulators [16,17,18] are over-dimensioned for a DRL agent that must execute many millions of steps. The aim of these existing simulators is not to be fast in terms of actions per second but rather to run close to real-time in robotics simulations, e.g., the UUV simulator [16] has a real time ratio of 0.75 on an Intel i7 processor (meaning that 0.75 s of real time are simulated in 1 s of execution time). Therefore, another contribution of the paper is the implementation of a fast 3D simulator that can easily perform enough steps per second to train DRL agents. The agents are trained using a set of environments created with the help of generative adversarial networks (GANs), themselves trained on real underwater surfaces.

### 1.3. Structure of the Paper

Next, Section 2 introduces the simulator, describes how it works and investigates its speed. Section 3, briefly discusses background on DRL. Section 4 presents the DRL application to mapping using the simulator, together with some mapping-specific DRL improvements. Section 5 gives the experiments and discusses their results. Section 6 concludes the paper and provides ideas for future work.

## 2. Simulator

The simulator consists of the environment, the representation of the map, the dynamics of the robot and the sensor model.

**Environment:** The map is represented as a grid. The map is loaded and created from a heat map image, where the temperature is the height of the environment. When creating the environment, the length, width and height of the grid are passed as arguments. Some of the areas can be marked as litter. The 3D map then contains the following information: 0=notoccupied, 1=occupied and 0.5=litter (Figure 1). A reduced-complexity 2.5D representation is also created, consisting of two matrices. One of these matrices, denoted by $\hat{H}$, stores the seafloor height for every 2D coordinate *i*, *j*. The other, denoted by *m*, is labeled with litter. The ground truth litter map is defined as follows: $m_{i,j}=0=surface$ and $m_{i,j}=1=litter$. After the 3D and 2.5D representations are created, the agent can be placed at any position in the environment.

To be able to use naturally-looking maps, the initial heat map images are created using a Generative Adversarial Network (GAN) [19] trained on on real data from the OpenTopology website. The dataset used to train the GAN is the Salton Sea Lidar Collection [20], which has a density of five points per square meter. This point cloud dataset was transformed into an image and the naturally-structured images (which do not show a flat surface but have some tree-like shapes) were chosen to train the GAN (Figure 2). The generated heatmaps were transformed back into a point cloud and finally into a voxelmap.

**Map representation:** The real map must be differentiated from the beliefs about the map. This difference is caused by limited range and errors in the sensor, see the sensor model below. In particular, the sensor can give different measurement outputs when measuring the same object from the same or a different pose. The recovery of a spatial world model from sensor data is best modeled as an estimation theory problem.

**Figure 1 sensors-22-05384-f001:**
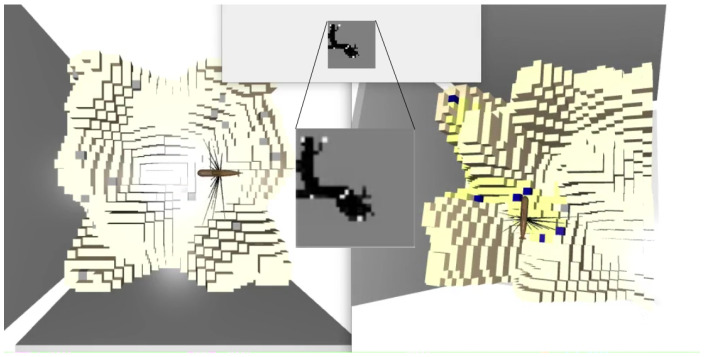
Left side: The ground truth map, where the gray areas are litter. Right side: Map discovered by the agent. A yellow color of the voxels means there is a strong probability for no litter, and blue means a high probability of litter. Top and center: 2D projection of the currently discovered map, where white represents litter, and black is no litter.

**Figure 2 sensors-22-05384-f002:**
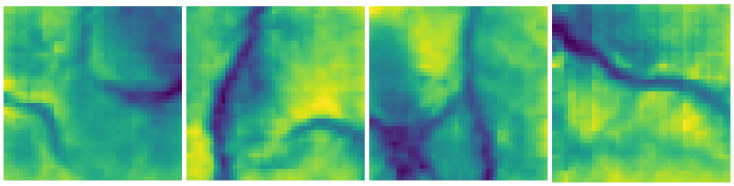
From the left: The first two images are real data from the OpenTopology website. The last two pictures are generated 2D images with GANs.

One of the most popular methods of probabilistically representing a map and merging new observations is through occupancy grids. This is due to some useful properties: it is easy to calculate an exact posterior distribution for each voxel of the map and to handle sensor noise. In the occupancy grid, the state variable $s(C_{i,j})$ associated with the cell $C_{i,j}$ is defined as a discrete variable with two possible values, occupied and empty. In practice, the probability that a cell is occupied is stored: $b_{i,j}=P\left[s\left(C_{i,j}\right)=o\right]$. Then, the probability that the cell empty is: $P\left[s\left(C_{i,j}\right)=e\right]=1-P\left[s\left(C_{i,j}\right)=o\right]$ [21].

**Robot state:** The pose P of the agent consists of a vector $P_{T}=\left[p_{x},p_{y},p_{z}\right]$ that stores the position of the agent and a rotation matrix $P_{R}$ that stores the attitude of the agent relative to the world coordinates.**Action and transitions:** The translation actions are performed in the agent’s coordinates. Before the agent performs an action, a check is made to see if the action is legal, i.e., whether the action would cause the agent to leave the environment or bump against the bottom. If the action is illegal, a collision signal is sent. The actions are defined here as follows:

Translate: forward, backward, left, right, up and down.Rotate: clock wise or counter clock wise around each axis.

A matrix A stores all possible actions. Each line in this matrix represents one possible action and defines either the length *d* (for translations) or the angle *r* (for rotations) of the action.
(1)$$A=\left[\begin{array}{cccccc} d & 0 & \cdots & \cdots & \cdots & 0\\ -d & 0 & \cdots & \cdots & \cdots & 0\\ 0 & d & 0 & \cdots & \cdots & 0\\ \vdots & -d & 0 & \cdots & \cdots & 0\\ \vdots & 0 & d & \cdots & \cdots & 0\\ \vdots & \vdots & -d & \cdots & \cdots & 0\\ \vdots & \vdots & 0 & r & 0 & 0\\ \vdots & \vdots & \vdots & -r & 0 & 0\\ \vdots & \vdots & \vdots & 0 & r & 0\\ \vdots & \vdots & \vdots & \vdots & -r & 0\\ \vdots & \vdots & \vdots & \vdots & 0 & r\\ 0 & 0 & 0 & 0 & 0 & -r \end{array}\right]$$

To calculate the new pose of the agent, when performing a transition the multiplication of the agent attitude $P_{R}$ with the vector of the performed action *a* is added to the current world coordinate pose $P_{T}$: (2)$$P_{T}^{\prime }=P_{T}+P_{R}\cdot a,$$
e.g., to move forward on the x axis, $a={\left[d,0,0\right]}^{T}$.

If a rotation action is performed, the chosen line from the action matrix A is transformed into a rotation change matrix $A_{R}$, and this change in the rotation is multiplied by the previous agent attitude matrix to get the new attitude:(3)$$P_{R}^{\prime }=P_{R}\cdot A_{R}$$

**Sensor model**: The sensor model is based on a multi-beam sonar where an overall angle $\hat{b}$ is covered by the angles of the combined *K* beams. Every beam has an opening angle $\hat{o}$, and this opening angle is represented by *L* rays, see Figure 3.

A 4-dimensional array $S\in R^{K\times L\times 3\times 3}$ is defined where the first dimension represents the beams *k*, the second dimension corresponds to the rays *l* of each beam *k*, and the last two dimensions represent the rotation of the ray relative to the agent.

The sensor array is initialized using a function $\hat{M}$, which takes an axis and an angle as input and gives a rotation matrix as output, which rotates with the input angle around the input axis. First, the rotation for each beam *k* is calculated, using the separation angle $\tilde{s}$ between beams:(4)$$\begin{array}{c} S_{k}^{beam}=\hat{M}\left(x,\tilde{k}\cdot \tilde{s}\right),\;\tilde{k}=-\frac{K-1}{2},\ldots ,\frac{K-1}{2} \end{array}$$
where $\tilde{k}$ gives the relative index of the beam, and the number of beams *K* has to be odd. Next, for each beam *k*, the individual ray rotations are computed:(5)$$\begin{array}{c} S_{k,l}^{ray}=S_{k}^{beam}\cdot \hat{M}\left(y,\tilde{l}\cdot \tilde{o}\right),\;\tilde{l}=-\frac{L-1}{2},\ldots ,\frac{L-1}{2} \end{array}$$
where $\tilde{o}$ is the opening angle divided by the number of rays, and *L* is the total number of rays per beam, which must be odd. $S_{k,l}^{ray}$ represents the ray rotation relative to the agent. When the agent changes its attitude or its position, the full sensor array is recalculated as follows:(6)$$S_{k,l,\cdot ,\cdot }=P_{R}\cdot S_{k,l}^{ray}$$
where subscript · means that all elements along that dimension are taken, so that $S_{k,l,\cdot ,\cdot }$ is the rotation matrix of ray k,l.

**Figure 3 sensors-22-05384-f003:**
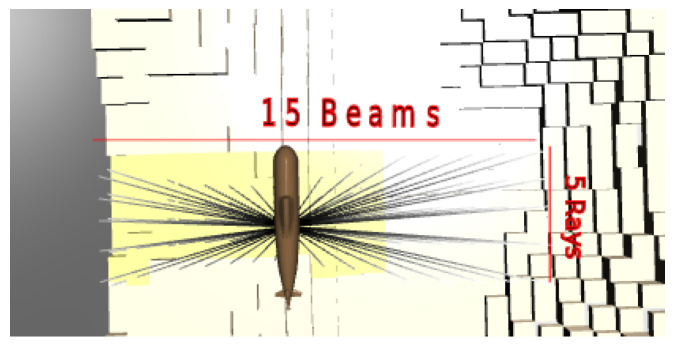
The simulator has 15 beams, each represented by five rays.

**Simulated sensor and map updates:** To read the sensor data the simulator checks if in the agent’s sensor range the voxels are occupied. A distance variable $a_{z}$ is defined in the ray’s z coordinate, which is increased in a loop with a step of $\frac{1}{u}$ voxels until the maximum sensor range is reached. The sensor rotation matrix from the 4D array, $S_{k,l,\cdot ,\cdot }$, is multiplied with a vector $\left[0,0,-a_{z}\right]$ to check if, at the current position along the ray, the voxel is occupied; if not, $a_{z}$ is increased. This continues in a loop until the sensor range is reached.

If the ray hits an occupied voxel, which represents either litter or the seabed, a measurement *z* is taken of the map *m* at positions i,j associated with that voxel (Figure 4). In addition, the simulator checks whether there is another voxel above this voxel (Figure 4c). If yes, than the surface voxel would not be reached, and thus the map would no longer be 2.5D. Instead, this voxel is ignored. Otherwise, the belief is updated and the current ray is marked as done (Figure 4a,b), so that it will not measure a second voxel. When a measurement is made, a distance $d_{k,l}$ is taken along the ray until the point that the voxel was hit. Note that Figure 4b shows an edge case where the ray goes “through” the upper left voxel and hits the lower right voxel.

Each of these rays should represent not only the aperture where the measurement is made but also different reflections of the signal; incorrect measurements are possible. The probability of a correct measurement of a given ray in all beams is calculated using an array C, which stores the values $C_{l}$ for the accuracy of a measurement, which simulates reflection depending on the angle of the opening. A decay factor ψ∈[0,1] is used to represent the decreasing accuracy of a measurement with the distance. The accuracy of the measurement for ray *l* of beam *k* is computed as follows:(7)$$c_{k,l}=\psi^{d_{k,l}}\cdot C_{l}$$
where $c_{k,l}\in \left[0,1\right]$.

The sensor makes erroneous measurements with the probability of $1-c_{k,l}$:(8)$$z_{k,l}=\left\{\begin{array}{cc} m_{i,j}, & \mathrm{with}\mathrm{probability}c_{k,l}\\ 1-m_{i,j}, & \mathrm{with}\mathrm{probability}1-c_{k,l} \end{array}\right.$$

Given measurement $z_{k,l}$, the belief is updated with the Bayesian rule:(9)$$b_{i,j}^{\prime }=\frac{P\left(z_{k,l}|m_{i,j}=1\right)\cdot b_{i,j}}{P\left(z_{k,l}|m_{i,j}=0\right)\left(1-b_{i,j}\right)+P\left(z_{k,l}|m_{i,j}=1\right)\cdot b_{i,j}}$$

A belief matrix B is defined that includes all elements $b_{i,j}$.

At each step, for each location i,j, the entropy *h* is calculated from the belief, as follows:(10)$$h_{i,j}=-\left(b_{i,j}\cdot \log\left(b_{i,j}\right)+\left(1-b_{i,j}\right)\cdot \log\left(1-b_{i,j}\right)\right)$$

Notation H is an entropy matrix that includes all elements $h_{i,j}$.

**Speed of the simulator:** The main motivation to build the simulator was speed. Figure 5 shows the dependency of the speed on the amount of rays and on the map size. In this figure, *u* is 3, and the ray length/range is 11. The tests were performed while choosing random agent actions, on a computer running Ubuntu 20.04 and having an Intel Core i7-8565U CPU and 16GiB RAM. The smallest speed is around 100 steps per second, which means 10,000 s (about 3 h) are needed to simulate one million samples. This is acceptable for DRL.

## 3. Background on DRL

Reinforcement learning deals with the problem of how to map situations (states *s*), of a learner, called an agent, to actions *a*to maximize a cumulative numerical reward signal. The agent is interacting with an environment as follows. The agent chooses actions, and the environment is responding to those actions by presenting new states to the agent and giving rewards *R*. The agent aims to maximize (in expected value) the sum of the discounted rewards over the future trajectory [22]:(11)$$G_{t}=R_{t+1}+\gamma R_{t+2}+\gamma^{2}R_{t+3}+\cdots =\sum_{k=0}^{\infty }\gamma^{k}R_{t+k+1}$$
where *G* is the return, *t* is the time step, and γ∈[0,1) is the discount factor.

If the agent is following policy π at time *t*, then π(a|s) is the probability that $A_{t}=a$ in state $S_{t}=s$, where a∈A for each s∈S. The value of taking action *a* in state *s* under a policy, $Q_{\pi }\left(s,a\right)$, is defined as the expected return starting from *s*, taking the action *a* and thereafter following policy π [22]:(12)$$Q_{\pi }\left(s,a\right):=E_{\pi }\left[G_{t}|S_{t}=s,A_{t}=a\right]$$

This Q-Value function satisfies a recursive relationship called the Bellman equation:(13)$$Q_{\pi }\left(s,a\right)=\sum_{s^{\prime }\in S}P_{ss^{\prime }}^{a}\left(r\left(s,a\right)+\gamma \cdot \sum_{a^{\prime }\in A}\pi \left(a^{\prime }|s^{\prime }\right)Q_{\pi }\left(s^{\prime },a^{\prime }\right)\right)$$
where $P_{ss^{\prime }}^{a}$ is the transition probability of landing in state $s^{\prime }$ when starting in state *s* and taking action *a*. Moreover, *r* is a reward function that generates the received reward *R* when taking action *a* in state *s*.

The optimal Q-value function $Q^{*}$ gives the best Q-values obtainable under any policy:(14)$$Q^{*}\left(s,a\right)=\max_{\pi }Q_{\pi }\left(s,a\right)$$
and satisfies the Bellman optimality equation:(15)$$Q^{*}\left(s,a\right)=\sum_{s^{\prime }\in S}P_{ss^{\prime }}^{a}\left(r\left(s,a\right)+\gamma \cdot \max_{a^{\prime }\in A}Q^{*}\left(s^{\prime },a^{\prime }\right)\right)$$

Once $Q^{*}$ is available, choosing actions that are greedy in this Q-function leads to a (deterministic) optimal policy, which achieves the objective:(16)$$\pi^{*}\left(s,a\right)=\left\{\begin{array}{cc} 1 & \mathrm{for}\mathrm{some}a\in \arg\max_{a^{\prime }}Q^{*}\left(s,a^{\prime }\right)\\ 0 & \mathrm{otherwise} \end{array}\right.$$

The temporal difference (TD) is an important concept in online RL. Denote the Q-function at step *t* by $Q_{t}$, and then, given an action $a_{t}$ in state $s_{t}$, the TD is defined as:(17)$$\delta_{t}=R_{t+1}+\gamma \max_{a^{\prime }}Q_{t}\left(s_{t+1},a^{\prime }\right)-Q_{t}\left(s_{t},a_{t}\right)$$
and can be understood as a sample of the difference between the right-hand side and the left-hand side of the Bellman optimality Equation (Equation 15), when applied to the current Q-function. To learn now the Q-values, they can be updated over time using the TD as an error signal:(18)$$Q_{t+1}\left(s_{t},a_{t}\right)=Q_{t}\left(s_{t},a_{t}\right)+\alpha \delta_{t}$$
where α is the learning rate. This algorithm is called Q-learning.

Consider next the problem of selecting the actions during learning. To solve it, one must address the so-called exploration–exploitation problem. Exploration allows the agent to collect new experiences to improve the accuracy of the estimated Q-values, whereas exploitation chooses the greedy action, thus, exploiting the agent’s current Q-value estimate. The epsilon-greedy strategy solves the exploration–exploitation dilemma in the following way. It chooses a random action $a_{t}$ at timestep *t* (explores) with probability ϵ and otherwise chooses the greedy action (exploits):(19)$$a_{t}=\left\{\begin{array}{cc} a\mathrm{uniformly}\mathrm{distributed}\mathrm{random}\mathrm{action}, & \mathrm{with}\mathrm{probability}\epsilon \\ \arg\max_{a}Q_{t}\left(s_{t},a\right), & \mathrm{with}\mathrm{probability}1-\epsilon \end{array}\right.$$

A widely used approach is to start with ϵ=1, and over time decrease it until close to 0.

**Deep reinforcement learning** uses deep neural networks (deep NNs) to estimate the Q values. Denote by Θ the parameters of the NN and by Q(s,a,Θ) the corresponding approximate Q-value function. In the case of the DDDQN [3] and Rainbow [4] algorithms that will be used, two networks are employed. The reason is that taking an action with the highest noisy Q-value in the maximum application within the TD, using the “normal” parameters Θ, would lead to overestimation of the Q-Values. To solve this, the algorithms compute the maximum in the TD using another set of parameters $\Theta^{+}$, which give the so-called target network [23]:(20)$$\delta^{+}=R_{t+1}+\gamma \max_{a^{\prime }}Q\left(s_{t+1},a^{\prime },\Theta^{+}\right)-Q\left(s_{t},a_{t},\Theta \right)$$

Usually, $\Theta^{+}$ is a delayed version of Θ, updated only from time to time.

To update Θ, observed transitions are stored in an experience replay buffer, and various forms of gradient descent are performed on a loss signal that involves the TD of so-called minibatches of transitions sampled from the buffer. The math is given in the specific case of prioritized experience replay (PER) [5], which is the concept of giving priority to transitions with a large TD error $\delta^{+}$, instead of sampling transitions uniformly randomly. The TD error intuitively indicates how “surprising” or wrong the prediction was, and the NN trains more often (with a higher priority) on the transitions, which was more surprising. The probability of sampling transition with index *i* is:(21)$$P\left(i\right)=\frac{p_{i}^{\rho }}{\sum_{k}p_{k}^{\rho }}$$
where $p_{i}>0$ is the priority of transition *i*. The prioritization is rank-based where $p_{i}=\frac{1}{\mathrm{rank}(i)}$, where rank(i) is the rank of transition *i* when the replay memory is sorted according to the TD magnitudes $|\delta_{i}^{+}|$. The exponent α determines how much prioritization is used, with ρ=0 corresponding to the uniform case.

Prioritizing transitions causes bias because it changes the distribution. To compensate for this bias, importance-sampling (IS) weights are used:(22)$$w_{i}={\left(\frac{1}{N}\frac{1}{P(i)}\right)}^{\hat{\beta }}$$
where *P* is the probability to sample a transition *i* and *N* is the current replay buffer size. IS is annealed by increasing $\hat{\beta }$ from ${\hat{\beta }}_{0}$ to 1, which means its effect is felt more strongly at the end of the learning. This is because the unbiased nature of the updates in DRL is most important near convergence [5].

Finally the loss function *l* for training the network can be defined. This loss function is computed over mini-batches B and consists of the weighted mean square TD error:(23)$$l=\frac{1}{|B|}\sum_{i\in B}w_{i}\delta_{i}^{2}$$

## 4. Application of DRL to Underwater Mapping

To apply DRL to the mapping problem, the state, actions and reward signals are first defined. Then, the methodological improvements made to the DRL algorithms are explained.

**State**: The state *s* of the agent is a tuple composed of the pose P, the belief B, the entropy H and the height $\hat{H}$: (24)$$s=\langle P,B,H,\hat{H}\rangle$$The state is normalized between −1 and 1 to help the neural network learn. To do so the pose P is split into $P_{T}$ (position) and $P_{R}$ (rotation):(25)$$s^{N}=\langle \begin{array}{c} P_{T}^{N}=\left(\frac{P_{T}}{{\hat{P}}_{max}}-0.5\right)\cdot 2,\\ P_{R}^{N}=\left(\frac{P_{R}}{2\pi }-0.5\right)\cdot 2,\\ B^{N}=\left(B-0.5\right)\cdot 2,\\ H^{N}=\left(-\frac{H}{log(0.5)}-0.5\right)\cdot 2,\\ {\hat{H}}_{N}=\left(\frac{\hat{H}}{{\hat{H}}_{max}}\right)-0.5)\cdot 2 \end{array}\rangle$$
where ${\hat{P}}_{max}$ is the size of the environment, ${\hat{H}}_{max}$ is the maximum height of the environment, and operations are applied element-wise.

This state is used to create the input of the neural network. In order to take advantage of the spatial dependency of the information, a convolutional neural network is used. However, since there is also a spatial dependency between the pose/attitude and the entropy/belief/height, the pose information is not connected with a dense layer after the convolution layers, as would be the classic idea. Instead, this information is added in two more channels in the first convolution layer, see Figure 6. In this way, the Neural Network can already make some connections between the position and the belief/entropy/height in the first layer of the NN.

The pose is passed as a 2D matrix (2.5D representation) where the normalized height of the agent is stored at the x,y coordinate of the agent. This works only when the agent’s position and actions are integers. If the pose or actions were continuous, all three values (x, y and height) would have to be added instead. The rotation matrix is centered at the x,y position of the underlying channel. To ensure the rotation matrix fits at all positions, the channel is padded.

**Actions:** The actions are those defined in the robot model of Section 2.**Reward:** The goal of the agent is to find the litter as fast as possible. In general, rewards can be defined based on the belief and the entropy of the map. To help the agent learn, rewards are provided both for finding litter and for exploring the map.

Due to the stochastic sensor model, if all the voxels were considered when computing the rewards, the variance would be large. To see why, assume $c_{k,l}=c$ for all k,l. Then, for one ray, the observation *z* is either 0 or 1, leading to high variance. For many rays, the count of the average of the rays that are correct tends to *c*. This makes belief updates more predictable on average, and hence rewards and Q-values have less variance. Many rays can thus reduce the variance but they are making the calculations slower.

Even if many rays are used, the agent could rotate so that only a few rays hits the voxels, and thus the variance of a measurement would increase again. This can lead to the fact that, even after converging to the real *Q* value, a high TD error is still observed due to the noise in the measurements. A second problem that appears due to variance is that the algorithm will prioritize from the experience replay buffer a measurement with a high variance, instead of a state with a “real” high error.

To reduce the variance without unduly increasing the number of rays, the rewards are defined so that the agent only receives a reward from a voxel once if the agent is (virtually) sure that it has identified this voxel as a litter (high value of the belief; component $R_{L}$) or that it has discovered the voxel (low value of the entropy; component $R_{H}$):(26)$$R_{L}=\frac{1}{l}\sum_{i,j}I\left(b_{i,j}^{\prime }\geq 0.999\wedge \left(i,j\right)\notin L\right)$$
(27)$$R_{H}=\frac{1}{h}\sum_{i,j}I\left(h_{i,j}^{\prime }\leq 0.001\wedge \left(i,j\right)\notin D\right)$$
(28)$$R_{C}=\left\{\begin{array}{cc} -1, & \mathrm{if}\mathrm{crash}\\ 0, & \mathrm{otherwise} \end{array}\right.$$
(29)$$R=R_{L}+R_{H}+R_{C}$$

Here, I is an indicator function (equal to 1 when the argument is true). M represents the 2.5D grid of voxels, and L⊆M represents voxels, which are labeled by the agent as litter. After receiving the reward $R_{L}$, the voxels are added to the set L. The same applies for the entropy, where D⊆M represents discovered voxels. After receiving the reward $R_{H}$ for these voxels, they are added to the set D. l and h are parameters for the reward function. $R_{C}$ is a collision penalty, when a collision is triggered by the environment. Note that reward computation uses the belief and entropy after the Bayes update ($b^{\prime }$ and $h^{\prime }$).

**Entropy-dependent exploration:** In this environment, states are similar at the beginning, e.g., for the first state of the trajectories, the entropy and belief are similar (uniform belief with high entropy). As the trajectories of the agent get longer and it discovers more of the map at various locations, the states become more unique. This poses an exploration problem. For this reason, instead of only exploring at the beginning and decreasing the linearly as in traditional DRL [24], exploration is also made dependent on the entropy left on the map:(30)$$\epsilon =\frac{\theta }{H(s)}$$
where θ is an exponentially decreasing parameter over time and H(s) is the sum of entropy over the whole map.**Modified PER:** Unlike classic DRL tasks (e.g., Atari games), here the agent receives nonzero rewards in almost every state. At the beginning of a trajectory, as the locations that are easy to discover are found, high Q-Values are seen, which then decrease progressively. As a consequence, the TD-error δ at the beginning of a trajectory is also greater, which increases the probability that PER chooses an early sample of a trajectory to train the NN, at the expense of later samples that may actually be more unique and, therefore, more relevant. To avoid this, the TD-error is normalized by the Q-value before using it for PER:(31)$$\delta_{N}=\frac{\left(R+\gamma \max_{a}Q^{\prime }\left(s^{\prime },a,\Theta^{+}\right)\right)-Q\left(s,a,\Theta \right)}{Q(s,a,\Theta )}$$**Collision-free exploration:** By default, the agent can collide with the seafloor or with the borders of the map (operating area). Any such collision leads to termination of the trajectory, with a poor return, which is hypothesized to discourage the agent from visiting again positions where collisions are possible. This means both that the Q-values are estimated poorly for such positions and, in the final application of the learned policy, that those positions are insufficiently visited to map them accurately.

To confirm this, a simpler 2D grid problem was solved with Rainbow, where the agent is represented as a blue pixel and the obstacles are colored white; see Figure 7. The goal is coverage, i.e., to reduce the entropy over the whole map. A simple sensor model is used where, with every visit of a cell, the entropy decreases by 80% in that cell. When the agent hits an obstacle or the border of the environment, the episode ends.

Figure 8 shows some images of the remaining entropy of the environments from Figure 7, after the agent executes its policy in each of those environments. Clearly, the agent struggles to visit positions that are prone to collisions. An extreme case is when the agent must traverse a long tunnel where collisions are possible at each step, as in Figure 7a.

To eliminate the imbalanced visiting frequencies of environment positions, the method will exploit the fact that the dynamics of the agent and the topography of the environment are known in advance. Therefore, when the agent chooses an action that would lead to a collision, the collision is predicted, and the action is forbidden. Instead, a random legal action is chosen.

However, naively adding only the legal-action transition to the replay buffer would be problematic, because then the agent cannot learn to avoid the illegal transition and, therefore, will often get stuck by repeatedly “attempting to collide”. To solve this, the illegal transition is also stored in the buffer, followed immediately by the legal transition, see Figure 9. Since the illegal transition is terminal, the next-state Q-value is always 0 irrespective of the actual next state, and thus the two transitions are in fact not connected during learning. Learning can therefore proceed correctly, without having to change the structure of the replay buffer.

Overall, this change should enable the agent to more often reach positions previously deemed too dangerous, while simultaneously lengthening the episodes, which makes them more informative.

## 5. Experiments and Discussion

The DRL agents were trained on 220 maps, generated with GANs (Section 2). These maps were also rotated and mirrored to obtain, in total, 1660 different maps. The validation dataset contains 40 maps that are based on real data and different from the training maps. The DRL maps are lower resolution and obtained by resizing the original maps.

A number of 63 voxels were labeled as litter in every map. The litter is placed with a higher probability in deeper regions so that the agent can learn the connection “valley = higher reward”. Note that uniformly distributed litter would lead to a less interesting map coverage problem. The litter is placed starting from the deepest layer until the predefined amount of 63 pieces of litter are placed. By layer, the probability of a voxel being litter is, starting from the deepest layer, 60%, 43%, 30%, 21%, 14%, 10% and 7% (see also Figure 10). The litter is resampled at each episode from the described distribution; therefore, maps are different in terms of litter placement even when the shape of the seafloor repeats.

It is important to note that training on a variety of different maps, with random litter positions—with each combination leading to a different MDP—followed by validation on a different set of such MDPs, is a nonstandard and challenging way of applying DRL.

As a baseline against which all DRL results are compared, a lawn mower (LM) algorithm was employed. This is an effective algorithm to solve a coverage problem, because it reduces efficiently the entropy over the full map. Moreover, since there are no crossings of the trajectory, no unnecessary steps are taken. The LM starts in the corner of the map (x = 0, y = 1) and goes once through the whole map, by moving forward until end of the environment, moving three steps south, running backwards until the end of the environment, moving three steps south and repeating until the end of the map.

During DRL training, in all cases, the improved exploration Formula (Equation 30) is used. The PER buffer stores 1.5 million transitions and to optimize the NN the Adam optimizer [25] is used. The agent is initialized in random initial positions on the map.

During validation, the initial position of the DRL agent is the middle of the map, since from there, it can move fast to other locations. For both the DRL and LM agents, the same seed is used for placing the litter on the 40 validation maps so that the comparison is performed on identical maps. Moreover, collisions are not permitted during validation. Instead, if the action given by DRL is illegal, a random legal action is chosen instead. The reason for this is to ensure that the trajectory always has enough steps to properly compare to the LM.

In the sequel, experiments will be performed with two different types of DRL agents: double dueling deep Q-networks (DDDQN) in Section 5.1 and Rainbow in Section 5.2.

### 5.1. DDDQN Results and Discussion

The DDDQN experiments are run with a relatively informative sensor with a large number of rays and relatively small maps of 27×27×24 voxels. Specifically, the agent’s sensor has 15 beams, each with 5 rays, with a decay factor ψ=0.99, and the ray accuracy array is C=[0.83,0.9,0.95,0.9,0.83]. The problem is further simplified so that the agent only has four translation actions (it moves in 2D in a single plane at altitude 14); rotation and moving up and down are disabled.

The Adam optimizer [25] has a small learning rate $\left(10^{-7}\right)$. The maximum length of an episode is 800 steps. The parameters for the reward function are l=10 and h=100. The network structure can be seen in Figure 11.

In what follows, we first verify whether the TD-error normalization (Equation 31) has benefits, in a simpler coverage problem. Then, once that is established, the full-blown experiment is run where the DDDQN agent is compared with the LM.

**Normalization of the TD error in PER:** For this experiment only, several simplifications are performed. Only discovered voxels are rewarded, and a penalty is given when the agent crashes, $R=R_{H}+R_{C}$, while the litter component $R_{L}$ is ignored, meaning that only a coverage problem is solved. Two agents are run, one with the normalized TD errors (Equation 31) and the other without, over the same amount of steps (10 million). Collisions are allowed during validation.

During the experiment, the episodes of the agent using normalized TD errors were increasing in length. During the time, that the classic agent needed to made 50,000 episodes, the normalized-TD agent performed 46,500 episodes. Furthermore, the amount of discovered voxels (that have less than 0.001 entropy left) was around 11% higher with the normalized TD. Classic PER discovered in average 421.82 voxels whereas with normalized TD errors the agent discovered 466.48 voxels. Note that, if the same amount of episodes were taken for both agents instead of the same amount of steps, the normalized agent would have a greater advantage.

**DDDQN versus LM:**Figure 12 shows the comparison between the LM and DDDQN agents, this time for the full litter-discovery problem. As shown in Table 1, the DRL agent finds around 56% of the litter, while the LM-agent finds 98% of the litter at the end of the trajectory. On the other hand, the DDDQN agent finds after 50 steps on average around 4% more litter. Moreover, the large difference in the variance is remarkable: in the worst trajectory, after 120 steps (half of the trajectory) the DDDQN agent found at least 11 L whereas the LM-agent found 0.

Note that Table 1 centralizes all the results obtained from now on, and it will be referenced repeatedly.

**Figure 12 sensors-22-05384-f012:**
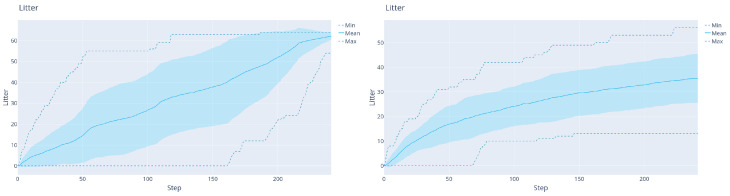
The results of the agents on the 40 validation maps, in the case with map size 27×27×24 maps. Left: LM agent. Right: DRL agent.

Overall, if the objective is to discover all the litter, the LM agent behaves better than DDDQN. By investigating, one reason was found to be that the DDDQN agent started with time to have a growing oscillation; thus, visiting the same regions many times without efficiently reducing the entropy. This is illustrated in Figure 13. Other possible reasons include an insufficiently powerful network or algorithm, which leads to the next section.

### 5.2. Rainbow Experiments and Discussion

As the DDDQN agent did not provide satisfactory results, several changes were made. First, the DRL algorithm was changed to a better one, called Rainbow [4]. Compared to DDDQN, Rainbow adds two key elements. (i) Instead of learning the expected Q-values, the agent learns a distribution over these Q-values [26].

This should help in mapping, since the noisy sensor and the litter distribution lead to significant stochasticity in the problem. (ii) To improve exploration, the last layer is modified to be a so-called noisy layer [27], without removing the custom exploration probability (Equation 30). Some retuning was needed after changing the algorithm. To prevent the PER from getting overly filled with states where the agent is oscillating between certain positions, the maximum trajectory length was reduced from 800 to 400 steps. The learning rate of the Adam optimizer was increased to $\left(10^{-6}\right)$. All agents were trained over 10 million steps and after 5 million steps exploration was disabled. Moreover, the final state was not labeled as “done” when the agent exceeds the maximum number of steps, which could also have caused problems during the DDDQN experiments.

Secondly, the network was made deeper, for reasons that will be detailed later. Deeper networks however tend to overfit and therefore to be unhelpful in DRL [28]. In computer vision, this problem is addressed with batch normalization, which was shown to smooth the optimization landscape, stabilizing gradient estimation [29]. Unfortunately, batch normalization causes instability in DRL, and instead an approach recently suggested by [30] is applied here: spectral normalization of network weights. Spectral normalization in DRL is most efficient on the largest fully connected layer, which is one after the last convolution layer. To handle the gradient vanishing problem, residual blocks [31] are added.

Thirdly, from time to time, some oscillations appear that lead to the agent getting stuck in a valley containing large quantities of litter. The impact of this during validation was reduced as follows. Consider the condition: in the last 22 steps, the agent revisited the same position more than five times. If this condition is encountered, the algorithm declares an oscillatory regime and, instead of applying the DRL policy, applies random actions until the condition becomes false. This simple change improves results by around 5%.

Finally, the mapping problem difficulty is increased, because—as confirmed towards the end of this section—even in the simpler problem from Section 5.1 better results are obtained with the new method. The size of the environment was increased from 27×27×24 to 32×32×24 voxels, and the sensor size was reduced from 15 beams × 5 rays to 15×3, with an accuracy array C=[0.9,0.95,0.9]. This poorer sensors makes it more difficult to find litter. The action space was increased from 4 to 8, and thus the agent can now also move diagonally in the plane.

The rest of the settings stay the same as for DDDQN.

In the remainder of this section, the following experiments are run. First, the baseline LM performance is reported for the new, more difficult problem. Then, since the reward function (Equation 25) is rather complicated, an attempt is made to simplify it, and it is shown that performance decreases. Similarly, to justify the use of the deep network, an experiment is run that shows performance is poorer with a typical, shallow DRL network. In the end, as a final improvement, collision-free training is performed using the method in Section 4. All results reported are on the 40 validation maps.

**LM baseline:** With the new configuration, the LM needs 350 steps to finish its trajectory. Figure 14 shows that on average 52.7 L items are found by the LM. During the first 31 steps, the agent does not find much litter. The reason for that is the poorer sensor, which must see again the same region to become sure that certain voxels are litter.

**Figure 14 sensors-22-05384-f014:**
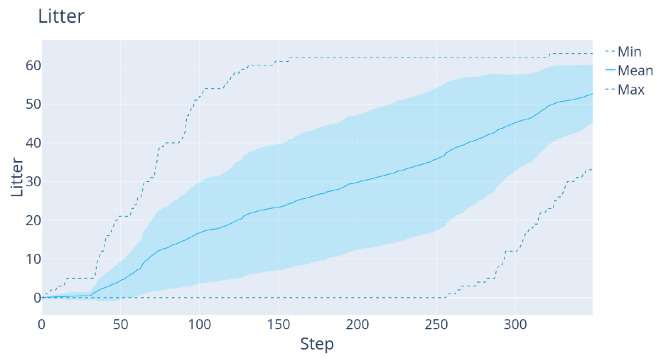
The results with the LM agent on 32×32×24 maps with the 15×3 sensor.

**The impact of entropy in the reward function:** An investigation was conducted to determine whether the entropy component is needed in the reward function. Figure 15a–c show results with no entropy component (corresponding to entropy parameter h→∞ in (Equation 25)), with h=100 (a low influence of entropy) and with h=10 (a larger influence of entropy). The litter parameter l is always 1.

When the rewards only include litter found, Figure 15a shows that the performance is good in the beginning but that the final amount of litter found is small, likely because the agent quickly becomes stuck, focusing only on the litter in a certain region. Figure 15b,c show that rewarding the reduction of entropy is beneficial, likely because entropy reduction pushes the agent to explore the map further.

**Figure 15 sensors-22-05384-f015:**
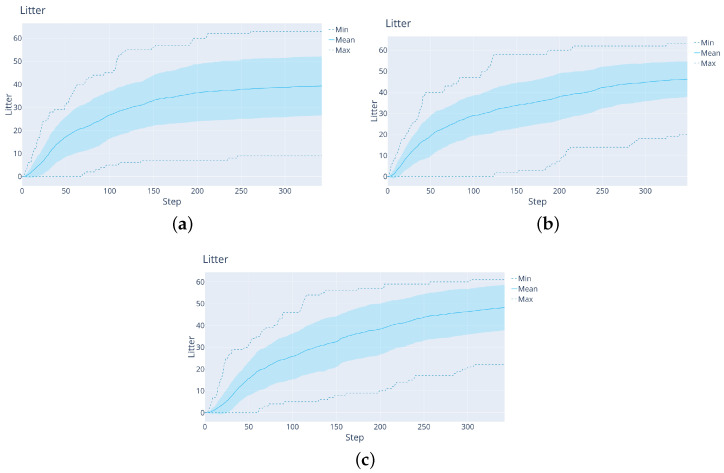
Rainbow agent performance for varying entropy influence on the rewards. From the left: (**a**) No entropy reward. (**b**) Entropy reward parameter h=100. (**c**) h=100.

As a rule of thumb for selecting the reward parameters l and h, a roughly equal ratio of litter and entropy rewards in (Equation 25) is recommended:(32)$$\frac{\frac{1}{l}\sum_{i,j}I\left(b_{i,j}^{\prime }\geq 0.999\wedge \left(i,j\right)\notin L\right)}{\frac{1}{h}\sum_{i,j}I\left(h_{i,j}^{\prime }\leq 0.001\wedge \left(i,j\right)\notin D\right)}\approx 1$$

The absolute magnitudes of l and h will likely depend on the map size and on the amount of litter.

**Deep versus shallow network:** To check whether the complexity of the network in Figure 16 is justified, here a shallower network is used, show in Figure 17. This network structure is appropriate for Atari games, the usual DRL benchmark. It has fewer layers with larger convolutional kernels. The results in Figure 18 and Table 1 show that this shallow network finds after 50 steps only 11.88 L on average and after 350 steps 36.78 L on average. The deeper network finds after 50 steps 19.3 L and after 350 steps 46.25 L.

**Figure 16 sensors-22-05384-f016:**
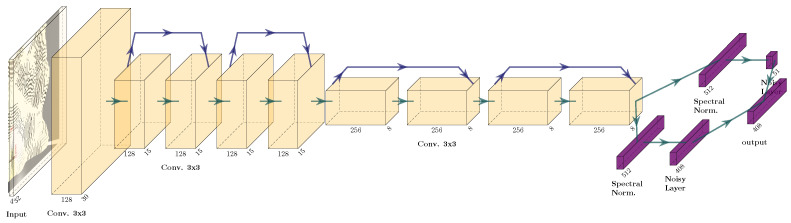
Network used for Rainbow.

**Figure 17 sensors-22-05384-f017:**
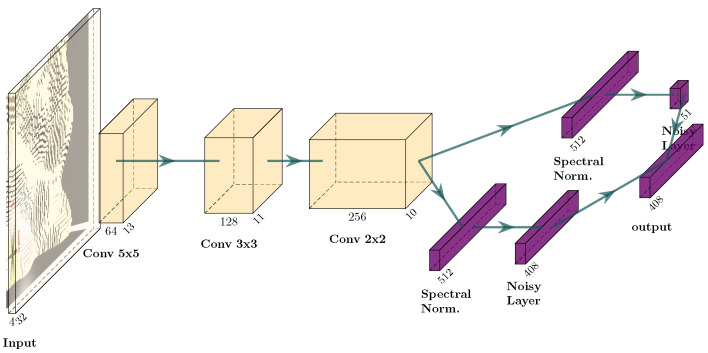
Shallow network structure with an input of size 4×32×32.

**Figure 18 sensors-22-05384-f018:**
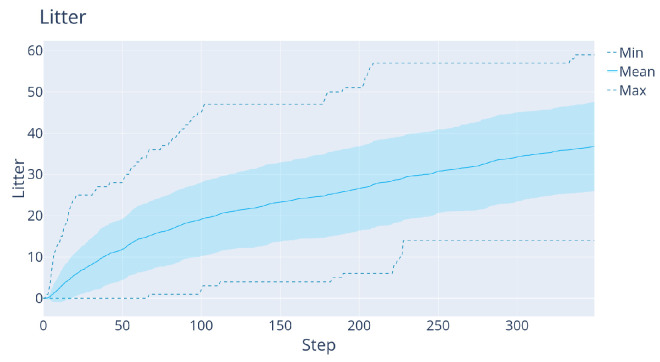
The results with the Rainbow agent and the shallow network. Here, h=100.

A question arises: why does a network structure widely used in DRL for Atari games not deliver good results in mapping? One explanation could be that in Atari games, images have a relatively simple structure (e.g., player ship and targets) and a few relatively simple feature detectors suffice to parse this structure. In contrast, in mapping, the seafloor profile, belief and entropy can all be arbitrarily complex surfaces. Therefore, it pays to have a larger amount of smaller convolutional kernels that are stacked in more layers.

To understand why smaller kernels may be beneficial, consider the case of a discrete kernel where the weights are 0 or 1. Then, with a 5×5 kernel the NN would need $2^{25}$ = 33,554,432 kernels to parse all possible configurations that may occur at the input of the first layer. In contrast, with 3×3 kernels only $2^{9}=512$ configurations exist; thus, fewer kernels will suffice. However, to identify larger-scale structure in the map, the smaller kernels must be stacked in more layers, which shows the need for a deeper network.

**Collision-free exploration:** For the final experiment, the following question was asked: could the collision and oscillation-avoidance measures, applied so far during validation, also help during training? Recall that, instead of simply avoiding collisions, the agent must additionally learn about them; therefore, the collision transitions are added to the PER as explained in Section 4. 

The results with this additional improvement are shown in Figure 19 and Table 1. Clearly, the latest Rainbow agent is always better than the LM: it both finds significantly more litter early on and increases the total amount of litter found. This is because the agent learns to exploit the prior that deeper regions contain more litter, see Figure 20 for a representative trajectory of this agent.

We also verified that this improvement holds for the simpler mapping problem used for DDDQN. Figure 19b shows the results, where again the collision-free Rainbow agent beats LM decisively early on, without a large drop in performance at the end. This justifies solving the increased-complexity task with Rainbow.

**Table 1 sensors-22-05384-t001:** Numerical comparison between all the agents, in terms of how fast they find litter early on (after 50 steps) and the total amount of litter found at the end of the trajectory. The smaller-map results are given above the double line, and the larger-map results below this line. The cyan background highlights the best agent for each type of map. In all the experiments the amount of false-positive findings were between 0.15 and 0.4 L items on average, i.e., very low.

Agent	50 Steps		End of Trajectory	
	Litter	%	Litter	%
LM	14.5	23%	62.0	98%
DDDQN l=10, h=100	16.9	27%	35.5	56%
Deep Rainbow l=1, h=25; no coll.	30.1	48%	56.4	90%
LM	4.5	7%	52.7	84%
Deep Rainbow l=1, no entropy (h→∞)	17.2	27%	39.3	62%
Deep Rainbow l=1, h=10	15.6	25%	48.2	77%
Deep Rainbow l=1, h=100	19.3	31%	46.3	73%
Shallow Rainbow l=1, h=100	11.9	19%	36.8	58%
Deep Rainbow l=1, h=25; no coll.	24.4	39%	55.4	88%

## 6. Conclusions and Future Work

### 6.1. Summary and Main Findings

This paper presented some first deep reinforcement learning results for underwater mapping. Several mapping-specific improvements were made to the DRL methods: a way to connect the pose of the agent efficiently in the convolution layer, a modified exploration schedule, normalization of the Q-values in prioritized experience replay and a technique to avoid collisions during learning while still learning from them.

To apply DRL, it was necessary to develop a new simulator for mapping with an autonomous underwater vehicle, which models a sonar-like sensor and is fast enough to provide the amount of samples required in a reasonable time.

After making the necessary improvements to a Rainbow DRL agent, it obtained decisively better performance than a lawn mower baseline. Moreover, experiments showed that a deeper network is useful in mapping (as opposed to a shallow network typically used in classical DRL) and that it is important to take entropy into account in the reward function.

### 6.2. Limitations and Future Work

To reduce the required number of samples for learning, future work could integrate all available model knowledge on the motion of the vehicle and focus learning only on the unknown part: the sensor model. This will also help to integrate any priors on the distribution of target objects that may be available prior to the experiment.

The agent and sensor models are currently quite simple, and it will be interesting to find more realistic models that are still computationally efficient enough to be useful for DRL. Along a similar line, it is important to see if the simplified-model DRL policies are applicable to more realistic mapping simulators or to a real system. The limitation to 2.5D maps means that objects in the water column cannot be mapped. Extending the framework to full 3D learning would be useful in practice but computationally challenging.

## Figures and Tables

**Figure 4 sensors-22-05384-f004:**
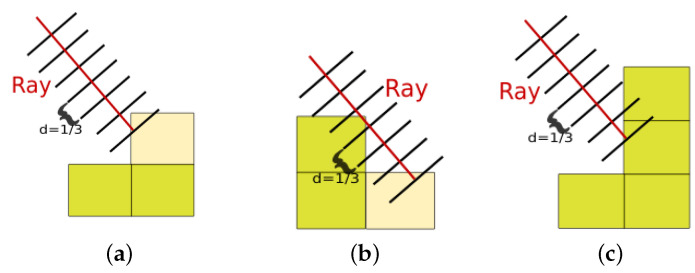
The lighter color visualizes which voxel is being measured. In the figures, the number of steps per voxel is u=3. From the left: (**a**) The belief of the upper voxel is updated. (**b**) Edge case: the ray is going through the upper voxel but the belief of the right lower voxel is updated. (**c**) No voxel is updated because the surface was not hit by a ray.

**Figure 5 sensors-22-05384-f005:**
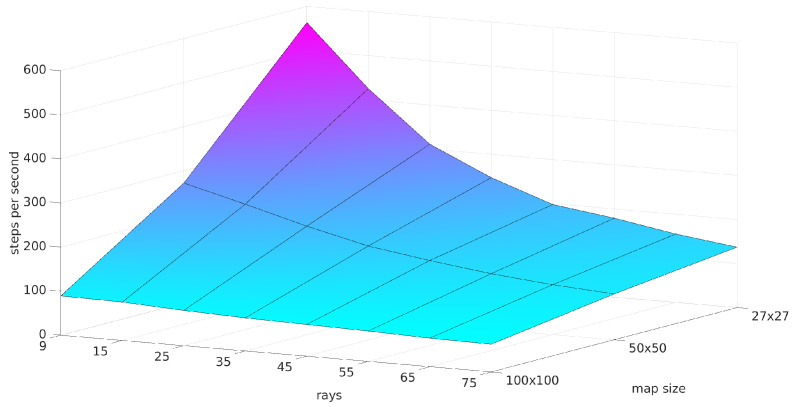
A graphical visualization of the simulator speed depending on the map size and on the number of rays. A sensor range of 11 was chosen with a number of steps per voxel of u=3.

**Figure 6 sensors-22-05384-f006:**
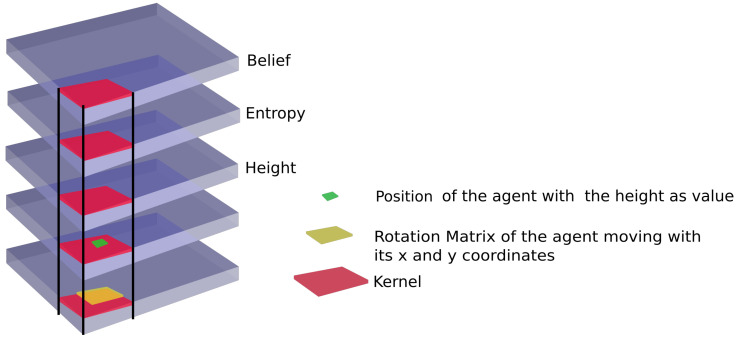
Representation of the state: The belief, height, entropy, position of the agent and attitude of the agent are each a 2D matrix. The picture illustrates how the kernel of the convolutional network connects the spatial information of the state already in the first layer.

**Figure 7 sensors-22-05384-f007:**
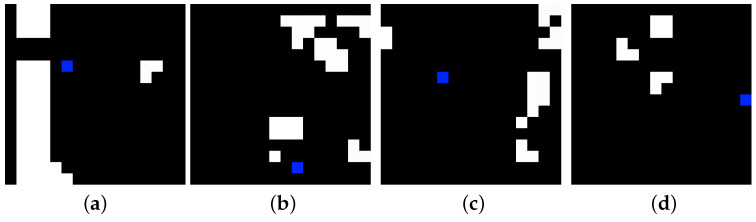
Examples of gridworld environments where the white color represents obstacles and the blue pixel is the agent. Each subfigure is a different environment.

**Figure 8 sensors-22-05384-f008:**
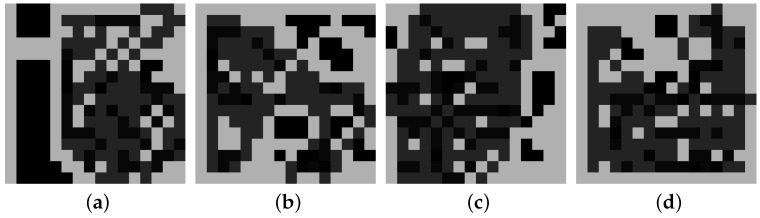
Solutions learned to a coverage problem for the environments above. The maps correspond one-to-one to those above. The lighter the gray color, the more entropy remains at this position at the end of the trajectory. Each subfigure is a different environment.

**Figure 9 sensors-22-05384-f009:**
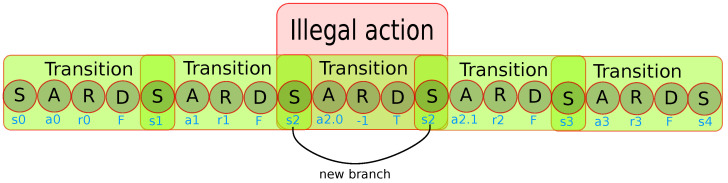
The structure of the PER for collision-free exploration. To save memory, every state is stored only once, linked to the next state in the trajectory. When a collision occurs, both the illegal transition $\left(s_{2},a_{2.0},r=-1,\mathrm{done}=\mathrm{True},s_{2}\right)$ is stored, and immediately after it, the legal transition that is actually performed in the environment, $\left(s_{2},a_{2.1},r2,\mathrm{done}=\mathrm{False},s_{3}\right)$. The dummy next state $s_{2}$ in the illegal transition is irrelevant, because the transition is terminal, and the next-state Q-values are always zero.

**Figure 10 sensors-22-05384-f010:**
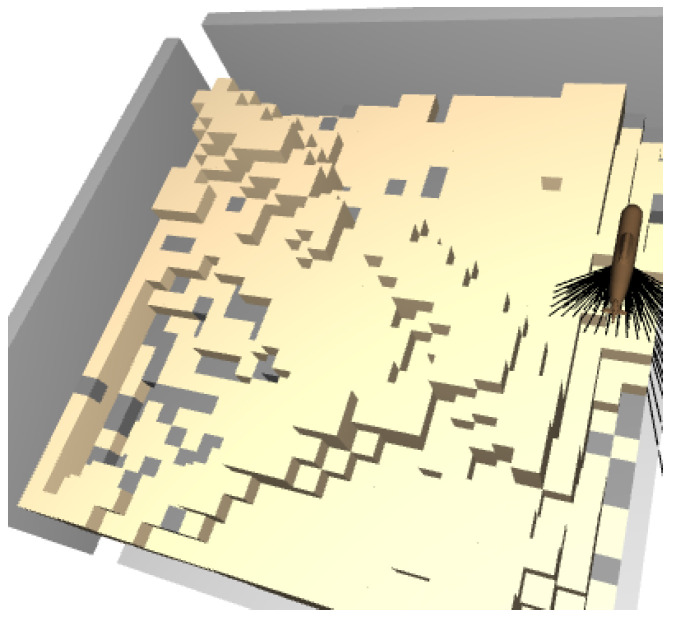
Distribution of litter for the experiment. More litter (gray voxels) was placed in the deeper layers (valleys) than in the higher layers.

**Figure 11 sensors-22-05384-f011:**
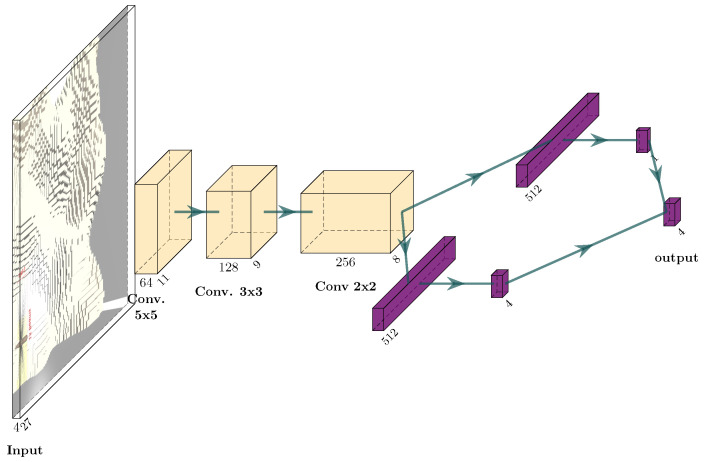
Network structure for the DDDQN experiments.

**Figure 13 sensors-22-05384-f013:**
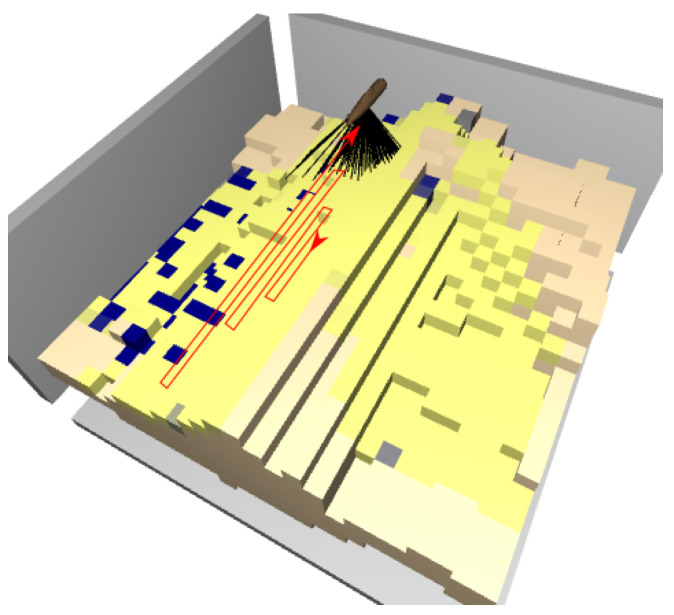
The agent starts to exhibit growing oscillation.

**Figure 19 sensors-22-05384-f019:**
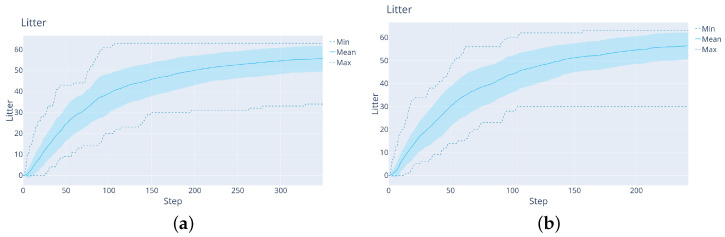
(**a**): The results of the collision-free Rainbow agent with the 3×15 sensor on 32×32×24 maps. (**b**): The results of a similar agent but with the 5×15 sensor on 27×27×24 maps (the version used for DDDQN).

**Figure 20 sensors-22-05384-f020:**
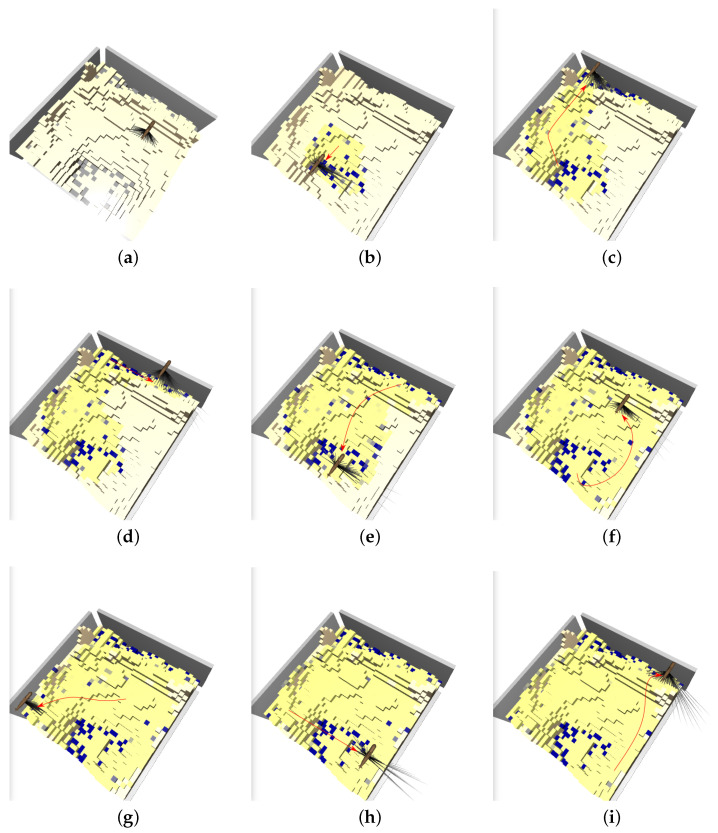
Visualization of the best agent’s trajectory using a sequence of snapshots (**b**–**i**). The first map (**a**) shows the ground truth (recall that litter is gray, low beliefs are yellow, and large beliefs are blue). The agents moves from the center directly to a valley where it finds large quantities of litter (**b**). Note that in (**c**), the agent does not take the shortest path to the next valley, likely because it is more efficient to approach the valley from one side instead of driving directly into the middle and then measuring parts of the valley twice.

## Data Availability

The set of maps generated with the GAN on which the agents are trained, see Figure 2 is available at: https://github.com/Mateus224/Mapping_Simulator/tree/main/Env/gan_images (accessed on 5 May 2022).

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
