# Peer review of "A Simulator and First Reinforcement Learning Results for Underwater Mapping"

_sensors, 2022, doi:10.3390/s22145384_

Round 1

Reviewer 1 Report

I have reviewed the manuscript entitled “A simulator and first reinforcement learning results for underwater litter mapping”. The researchers give interesting information to the readers. The manuscript is well-designed and well-described, but it needs some major corrections that should be considered to improve the manuscript.

  1. Modify the abstract. The first sentence should motivate your study, then explain clearly and concisely what you did and at the end your main findings and its importance/impact.
  2. Keywords are not interesting and should not repetition with title.
  3. I suggest you update your reference list by including more studies published over the last five years. For sure there are a lot of studies out there on this topic that can be used to explain the rationale of your study in the introduction and to discuss your results
  4. Literature review section need to add in order to judge novelty of your study.
  5. Surprisingly, there is no real discussion of the results
  6. How do your results compared to previous studies and how relevant are the findings?
  7. The conclusions should focus on the summary of the study, main findings, and possible implication
  8. Add ‘limitation of the study’ section after conclusion

Thank you.

Reviewer 2 Report

The topic is interesting and valuable, however it is not presented well. 

1. The quality of the language need to be greatly improved. For example, they use a number of "we" in the statements which can be changed.

2. In the introduction section, I hope the problems can be presented clearly. And the other sections off the paper should emphasize on solving these problems.  

3. Is table 1 necessary?

4. More references should be cited in the paper.

5. The full names of some abbreviations should be given when they first-time appear in the paper.  

Reviewer 3 Report

1- The abstract section contains a level of motivation that can attract the attention of the reader. However, the contributions from the methodological point of view should be emphasized more clearly. On the other hand, a sentence or two about performance would be helpful.
2- The Introduction section should be expanded. In particular, information such as open problems and resolved issues in the relevant literature should be presented. A systematic literature review would be helpful.
3- It would be useful to see the contributions presented in this study as articles.
4- More experimental analysis and evidence is needed. Comparative analysis and tables are also required.

Reviewer 4 Report

The submitted article describes an attempt to develop an automated search route-planning using deep reinforcement learning. The application motivating the work is attempting to find seafloor litter, but this paper presents a theoretical model that could be used for any target. As such, I recommend to edit the title. 

The manuscript was a bit hard to follow. There is a lot of text about the simulator. Even if design of this mapping simulator as a control module for the virtual robot took considerable effort, I don't think it is particularly novel. It is very efficient but there's not anything presented to benchmark it against, and I'm not a robotics software engineer. The GAN models come from data from the OpenTopography (not OpenTopology) archive. The background on RL is fairly basic and a good introduction to this topic. The DRL models were much more interesting to me. The experiment design, or at least the rationale for some of their choices was interesting. I would like to see more justfication for the reward system for instance. I also think that the TD-error introduced some odd behavior that influenced the outcome of the mapping; however, it's not possible to really assess that with the information given. 

In the end, their experiments did not turn out well. I was a bit surprised by how poorly the DRL performed in Figure 9. I think that there was something about the reward system that set this up for failure, although initially the results seem promising. In short, this study had interesting possibilities, was presented in a way that I'm not sure I could replicate (scientific soundness means it's repeatable), and with somewhat disappointing results that were not explained well. I recommend revision before publication.

Round 2

Reviewer 2 Report

The article can be accepted for publishing.

Reviewer 4 Report

The revised manuscript addresses all of my concerns in the previous version. I recommend publication of this revised version.